# Virgin Coconut Oil: A Dietary Intervention for Dyslipidaemia in Patients with Diabetes Mellitus

**DOI:** 10.3390/nu15030564

**Published:** 2023-01-21

**Authors:** Andina Setyawati, Moh Syafar Sangkala, Silvia Malasari, Nuurhidayat Jafar, Elly L Sjattar, Syahrul Syahrul, Haerani Rasyid

**Affiliations:** 1Lecturer of Medical and Surgical Nursing Department of Nursing Faculty, Universitas Hasanuddin, Makassar 90245, Indonesia; 2Lecturer of Community, Family, and Gerontology Nursing Department of Nursing Faculty, Universitas Hasanuddin, Makassar 90245, Indonesia; 3Lecturer of Medicine Faculty, Universitas Hasanuddin, Makassar 90245, Indonesia

**Keywords:** virgin coconut oil, nutraceutical, dietary supplements, hyperlipidaemia, diabetes mellitus

## Abstract

Hyperlipidaemia is causally related to coronary artery diseases (CAD) and peripheral artery diseases (PAD) in people with Diabetes Mellitus (DM). An in vivo study confirmed that virgin coconut oil (VCO) could maintain levels of lipids in the blood as effectively as conventional therapy. Therefore, this study aimed to determine the effect of VCO on the lipid profiles and ankle–brachial index (ABI) of patients with DM. In this experimental study with pre- and post-test design and a control group, the participants were selected purposively. The ABI was evaluated on the first visit. Baseline lipid profile readings were taken. Each participant took 1.2 mL/kgBW of VCO daily and divided it into three doses. After 30 days of taking VCO, laboratory examinations and ABI were repeated, and adverse events were evaluated. The dependent *t*-test and Wilcoxon sign rank test with a significance level of α ≤ 0.05 showed a significant decrease in low-density lipoprotein (LDL) (*p* = 0.002), a significant increase in high-density lipoprotein (HDL) levels (*p* = 0.031), a significant decrease in energy intake (*p* = 0.046) and cholesterol intake (*p* = 0.023) at the endpoint in the VCO group. In conclusion, this therapy is beneficial for maintaining lipid profile when combined with dietary therapy. Future studies should investigate the duration and dosage of VCO on patients to maintain lipid-linked protein.

## 1. Introduction

Along with the increasing welfare of people in developing countries, the incidence of degenerative diseases is also increasing, one of which is diabetes mellitus (DM). Globally, increasing DM prevalence requires attention, especially among health practitioners. Without intervention to halt these increases, there will be at least 629 million people with DM by 2045 [1].

With the high prevalence of diabetes, the incidence of complications, such as macrovascular and microvascular complications, is increasing. Macrovascular complications often have a poor prognosis and contribute to mortality, whereas higher numbers of death are caused by coronary artery disease (CAD) [2]. Lipid fractions of low-density lipoprotein (LDL), high-density lipoprotein (HDL) and possibly triglyceride (TG) are causally related to CAD in people with DM [3].

Recent study findings identified that LDL elevation and HDL reduction in people with DM are also strongly related to the reduction in the increasing incidence and future risk of peripheral artery diseases (PADs) [4]. North America had the highest reported prevalence of PAD globally, with the major populations being people with type 2 DM (T2DM) [5]. People with DM with high values of ankle–brachial index (ABI) tend to experience PAD and atherogenic lipid profiles. Furthermore, the characteristic of lipids was associated with ABI [6].

As the characteristic features of diabetic dyslipidaemia are a major risk factor for DM complication [7], maintaining good lipid profiles can prevent the development and progression of further hyperlipidaemia-related complications among people with DM. Treating the interplay of circulating lipids and the risk of CAD and PAD in people with DM is an emerging and important public health issue. Dyslipidaemia therapy has been demonstrated to provide more benefits to people with DM compared with those without DM, as people with DM responded faster [8].

Virgin coconut oil (VCO) is a dietary supplement that has been shown to improve lipid profiles in dyslipidaemia diabetic rats by decreasing the total cholesterol (TC) and LDL [9]. An in vivo study confirmed that VCO could reduce the TG levels in the blood and be as effective as conventional therapy using simvastatin [10]. In a higher-level study, the intake of 30 mL of VCO daily increased HDL levels in healthy people [11]. Another clinical study showed that VCO taken daily for 8 weeks reduced TC and TG levels in women with obesity [12]. The biological efficacy of VCO is due to its high polyphenol content such as ferulic acid, vanillic acid, *p-*coumaric acid, syringic acid, and caffeic acid. These polyphenols are reported to have beneficial effects on lipid parameters by reducing lipogenesis and enhancing the rate of fatty acid catabolism [13]. VCO is also reported to prevent and correct dyslipidaemia through regulating oxidative stress by reducing superoxide dismutase (SOD). The presence of polyphenols in VCO as bioactive molecules might have protected cell constituents against oxidative damage by directly reacting with free radicals in the extracellular environment or by upregulating the cellular antioxidant defence systems [14].

The therapeutic benefits of VCO have been attributed to its high polyphenol contents, such as ferulic acid, quercetin and caffeic acid [15]. The screened secondary metabolites of VCO that contribute to lipid profile maintenance are polyphenols. Polyphenols isolated from VCO were reported to prevent lipid abnormalities in rats by improving the antioxidant defence system [16]. Several studies have identified the pharmacologic effects of VCO in diabetic conditions. VCO demonstrates antidiabetic activity by reducing blood glucose levels, reducing glycated haemoglobin and regenerating damaged islet in diabetic rats [17,18,19]. In addition, VCO was found to maintain lipid profile in diabetic rats by reducing TC, TG and LDL levels and increasing HDL levels [20]. In addition, VCO therapy exerts a probiotic effect on diabetic conditions [21].

To our knowledge, no clinical studies have implemented VCO therapy to correct dyslipidaemia among people with DM; thus, studies investigating the beneficial potential of VCO therapy in people with DM and dyslipidaemia are novel. Therefore, this study aimed to investigate the effect of VCO on serum lipid profiles including TC, LDL, TG and HDL, and on ABI.

## 2. Materials and Methods

### 2.1. Materials

VCO El Medinah (P-IRT NO. 2073402011960-20) was used as a nutraceutical food supplement. TC and TG levels were measured using commercial assay kits (with sensitivity 3 mg/dL [0.08 mmol/L]) (Pars Azmoon Co., Tehran, Iran) with glycerol oxidation and cholesterol oxidation enzymatic methods by an auto-analyser (Echo plus Company, Italy). The HDL level was measured after the precipitation of beta-lipoproteins by dextran sulphate and chloride magnesium using the oxidation cholesterol method and an auto-analyser (with sensitivity 1 mg/dL). The LDL level was calculated using Friedewald’s formula as seen below [22]:

LDL = TC − HDL − (TG/5)

An 8 MHz Doppler (Parks Medical Elektronics, Inc., Aloha, OR, USA) was used to detect the ABI values.

### 2.2. Participants

The participants were recruited on an enrolment basis following the inclusion criteria. Participants were diagnosed with DM and dyslipidaemia, aged ≥20 years, regardless of sex, were visiting the endocrinology clinic on the day of enrolment and had normal body mass index. Patients with a history of routine VCO consumption, cardiovascular and renal diseases, taking statins or any-related medications, not completing VCO therapy at the proposed time, active smokers or experiencing adverse events (e.g., diarrhoea) were excluded. All eligible patients were asked to provide consent before participating in the study.

The calculated sample size was 142 (71 each group) computed based on the following formula [23,24]:$$n=\frac{\sigma^{2}\left[Z_{1-\alpha +Z_{1-\beta }}\right]2}{\mu_{0-\mu_{a}}}$$
where:

*n* = sample size;

*µ*_0_ = mean from baseline of total cholesterol = 4.66;

*µ*_a_ = mean from endpoint of total cholesterol = 3.94;

*Z*1 − *α*/2 = 95% confidence interval, alpha 0.05 this is 1.96;

*Z*1 − *β* = This depends on power, for 80% this is 0.84;

*σ* = standard deviation = 2.16.

The numbers recruited were 147 (73 in the intervention group and 72 in the control group). The consecutive sampling method was used. Individuals were randomly allocated into the intervention and control groups. The numbers finishing the study were 136 (68 in the intervention group and 68 in the control group). The reasons for leaving the study in the intervention group were diarrhoea (*n* = 2), nauseous (*n* = 1) and taste intolerance (*n* = 2). Four (4) participants in the control group were uncontactable at the end of the study.

### 2.3. Methods

The study was conducted in Wahidin Sudirohusodo and the teaching hospital of Hasanuddin University in Makassar, Indonesia. This was a quasi-experimental study using pre- and post-test with a control group.

After obtaining consent, the demographic data of the participants were collected. Serum TC, TG, LDL and HDL levels and ABI were measured at baseline (day 1) and endpoint (day 30). Participants were asked to fill in a dietary sheet daily. The participants were referred to a laboratory for lipid profile assessment. After 12 h of fasting, 5 mL of venous blood was taken for serum TC, TG, LDL and HDL measurements.

ABI was derived from the systolic blood pressure (BP) measured on the arm and leg after 10 min of rest in the supine position, with the arms and legs straight and at rest. Manual cuffs were used for all BP measurements, and arm circumference was determined during screening to select the appropriate cuff size. The same cuff size was used for the lower leg, and the straight wrapping technique was employed. Arm BP was measured using a sphygmomanometer and an 8 MHz Doppler to detect pulses. One measurement was made at each of the six sites in the following order: left arm, left ankle (dorsalis pedis and posterior tibialis), right arm and right ankle. The right ABI was calculated as the ratio of the higher right ankle pressure (dorsalis pedis or posterior tibialis) divided by the higher brachial pressure (right or left side). The left ABI was calculated using a similar method. The lower ratio of either side was considered the participant’s ABI [25].

Participants were asked to continue their lifestyle as usual. The VCO dosage was based on the participants’ weight. For days 1–3, the VCO dosage was 0.6 mL per kilogram bodyweight (kgBW) and was increased to 1.2 mL/kgBW for days 4–30. The calculated VCO was incorporated into meals at breakfast, lunch and dinner or snacks. This VCO contains lauric acid (47.96%), myristic acid (20.26%), palmitic acid (9.08%), acprylic acid (6.99%), capric acid (6.49%), oleic acid (5.18%), stearic acid (3.11%) and linoleic acid (0.81%). The researchers conducted telemonitoring to remind and assure VCO consumption schedules. Signs and symptoms were monitored and recorded on a daily basis. NutriSurvey Indonesia version 2017 was used for the recording of the daily food intake.

### 2.4. Statistical Analysis

Statistical analysis was done using IBM SPSS statistics 20 windows (SPSS Inc., Chicago, IL, USA). The results are given in mean ± standard deviation and for categorical data as a percentage. The absolute changes from baseline in the lipid parameters and ABI were computed between the two measurements, i.e., the baseline vs. end of the intervention by using paired *t*-test. The differences were compared between the two groups using the independent *t*-test. To test the hypothesis, a significance level of 0.05 was used to determine if the difference between the two groups was statistically significant.

The study was conducted according to the guidelines of the Declarations of Helsinki. The research proposal and all the procedures involving research study participants were approved by the ethics committee of the Faculty of Medicine, Hasanuddin University (KE/FK/1213/EC/2019).

## 3. Results

A total of 136 participants finished the study, i.e., 68 participants of the VCO group and 68 participants of the control group. As shown in Table 1, participants from the VCO group were older than the control group and most participants in all groups were male and had a BMI category as overweight. With further analysis, most participants consumed high amounts of carbohydrates (>26% of energy consumption) and cholesterols daily. The carbohydrate requirement per 24 h in healthy people is 30 times the body weight (unit calories), whereas that of patients with DM is 60% of the calculated carbohydrate amount in healthy people. Data were obtained by converting grams into calories (1 g = 4 calories).

The biochemistry data, i.e., total cholesterol (TC), triglyceride (TG), HDL, and LDL were not significantly different at baseline between the two groups as well as ABI values. But, the VCO group consumed significantly higher energy intakes than the control group at baseline.

The mean TC of the participants in the VCO and Control groups at baseline were 247.69 and 231.85 mg/dL, respectively (Table 1). After the intervention, there was no significant decrease in the TC of the participants enrolled in both groups (Table 2). However, the range of change from baseline to end point between groups was significantly different (*p* = 0.018), where the VCO group experienced a decrease in TC at 21.08 mg/dL while the control group experienced an increase in TC at 7.99 mg/dL (Figure 1). Similarly, The TG of the VCO and control group had no significant decrease after the intervention but a significant difference in a range of change between groups was observed (*p =* 0.039). The VCO group experienced a decrease in TG at 9.23 mg/dL while the control group experienced an increase in TC at 7.62 mg/dL (Figure 1).

The HDL in the VCO group increased significantly after the intervention. The mean difference of HDL was significantly different between groups (*p =* 0.010) as the VCO group experienced an increase in HDL at 9.4 mg/dL while the control group experienced a decrease in HDL at 6.15 mg/dL (Figure 1). The mean LDL of the VCO and Control groups at baseline were 179.62 and 156.31 mg/dL, respectively. After the intervention, there was a significant decrease in the VCO group. The mean difference of LDL in the VCO group (38.16 mg/dL) was significantly higher (*p =* 0.0001) than in the Control group (3.16 mg/dL) (Figure 1).

The mean energy intake in the VCO and Control groups were 1768 kcal/day and 1452 kcal/day at baseline, respectively. After the intervention, only the VCO group had a significant decrease in energy intake. The mean difference of energy intake in the VCO group (−414 kcal/day) was significantly lower (*p =* 0.0001) than in the control group (+26 kcal/day). A significant decrease in cholesterol intake was evident in the intervention group towards the end of the intervention (mean difference 168.30 mg, *p =* 0.023) (Table 2). For the other parameters, no significant difference in the ABI, carbohydrate intake, and IMT was observed.

## 4. Discussion

This study is the first study to have evaluated the effect of VCO on people with DM on dyslipidaemia. Overall, the VCO group showed more rapid effects from dyslipidaemia by showing a significantly higher reduction in the mean TC, TG and LDL levels and elevation in the mean HDL level than the control group after 30 days. These observed effects of VCO in the blood levels of the lipid profile in diabetic patients suggested that the extract is effective in ameliorating diabetic complications such as cardiovascular disease, arteriosclerosis, coronary artery disease, etc., due to its high content of polyphenols which are bioactive components in the extract [26]. Additionally, a decrease in energy intake and cholesterol intake was observed in the VCO group. However, the TC and TG levels in the VCO group decreased with no significant difference. Following the result, VCO has no effect on ABI. In line with the study findings, there was a relationship between TG and ABI in a previous study [27].

A low ABI value reflects atherosclerosis in the vessel wall, resulting in a decrease in perfusion and arterial circulation to the distal extremities. In a recent study, ABI value is associated with different risks of diabetes. Further, there is a relationship between TG and ABI. High amounts of TG affect blood vessel and cause atherosclerosis that contributes to the ABI value [27]. This explanation is in line with our study findings that VCO slightly decreased TG with no significant mean difference. Therefore, a decrease in ABI was not observed.

VCO evidently had anti-hyperlipidaemia activities. An in vivo study showed that applying VCO orally could decrease all lipid profiles, including TC, TG, HDL and LDL close to normal levels [14,28]. However, a recent clinical study showed that VCO increases HDL without changes in LDL [29]. Different from our study, this study had a population of cardiovascular diseases and long-term VCO taken orally at 4 g/day without considering patients’ weight.

In [26], although the TC and TG levels in the VCO group decreased with no significant difference within the VCO group, VCO improved the lipid profile by significantly decreasing LDL and increasing HDL levels. HDL is a free radical scavenger that can prevent beta-lipoprotein peroxidation. Therefore, lowering the HDL level is considered to prevent heart disease development in people with DM [30]. Epidemiologic evidence shows that when HDL levels are increased by 0.025 mmol/L, the associated risk of heart disease development is reduced by 2–3% [31]. It means that a 9.4 mg/dL increase in HDL after taking VCO in our study could decrease 42–36% of heart disease risk. Showing that there was a significant difference in means between VCO and control groups in all lipid profiles indicates that VCO has the potential to alleviate dyslipidaemia. Future research needs to optimise the dosage and frequency of use to lead to the preeminent effect. Another study found that taking VCO (15 mL twice daily) as a supplement for young healthy people increases HDL-C levels and there are no adverse effects, thus its use is recommended to reduce the risk cardiovascular problem [11]. Additionally, coconut, when it is administered in an isoenergetic balanced diet, could raise HDL cholesterol levels and decrease the TC/HDL cholesterol ratio in obese men [32].

VCO is also high in saturated fat (MCFA), which increases HDL-cholesterol levels compared with other plant-based oils that have polyunsaturated and monounsaturated fats [11]. MCFA upregulates apoA-1, the major protein component of HDL synthesised by the intestinal, which interacts with the ATP-binding cassette transporter A1 (ABCA1). This interaction mediates the efflux of cellular cholesterol to lipid-free apoA-1, thereby inducing reverse cholesterol transport and biogenesis of HDL particles. Therefore, MCFA improves the hepatic mRNA expression of ABCA1, which plays a pivotal role in HDL particle biogenesis [33]. Through another mechanism, VCO polyphenols activate the lecithin cholesterol acyltransferase enzyme in the endogenous pathway. As a result, the bonding process of cholesterol esters–lipoprotein cores improves; thus, low-density lipoprotein (VLDL) catabolism increases. The process means that free cholesterol and phospholipids resulting from VLDL breakdown are transferred to HDLs; thereby, HDL formation increases [34].

The significant decrease in the mean LDL level after VCO supplementation in our study could be due to a greater VLDL lipolysis and uptake rate through peroxisome proliferator-activated receptor (PPAR) activation. PPAR activation increases LPL lipolysis and increases the liver uptake of LDL. The lack of increase in LDL level is also explained as follows: the active chain elongation process might slow the VLDL production and MCFA downregulates the apoB expression. As a result, VLDL production decreased [35].

Other components of VCO that contribute to alleviating dyslipidaemia levels are polyphenols. In a systematic review, polyphenols were reported to improve LDL levels by ameliorating 25-hydroxycholesterol 7-alpha-hydroxylase (CYP7A1), a protein-coding gene that catalyses many reactions in the synthesis of cholesterol and other lipids. Multiple pathways may be involved in modulating these activities, including regulation through NF-Kβ/ERK and SIRT-RXR-FXR (LXR) signalling pathways, modification of circadian rhythm-associated genes, reversing cholesterol transport and bile salt hydrolysis. Several micro RNAs were also reported to target CYP7A1, and some of these are modulated by polyphenol supplementation [36].

In relation to ABI improvement, VCO reduces LDL levels by dissolving cholesterol. Thus, blood circulation becomes smooth, maintaining the ABI value [34]. The evidence is similar to our finding that the VCO group experienced refinement of the ABI value, whereas the control group experienced worsening of the ABI value. Unfortunately, the literature discussing further details of this mechanism is still limited. The trends in a reduction in the levels of TC, TG and ABI, although not significant in the VCO group, are sufficient reasons to claim that VCO exerts a protective effect on DM management in addition to preventing cardiovascular diseases as a DM complication. HDL serves as a ‘leader’ to other lipids because HDL reverses cholesterol transport from extrahepatic tissues to the liver for onward excretion in bile, thus reducing plasma cholesterol levels [37].

The VCO intervention also has a beneficial effect on diets, i.e., energy intake and cholesterol intake, but has no effect on carbohydrate intake and BMI. A study revealed that VCO have suppressive effects on hunger and desire to eat in normal-weight men [38]. As most of our participants were overweight men, this result does not match the effect on carbohydrate intake. However, another study reported that after 80 min of breakfast consumption, carbohydrate oxidation started to reduce, and an increase in fat oxidation after VCO breakfast seemed to occur. The inclusion of carbohydrate sources into breakfasts containing VCO could change substrate oxidation profile, once carbohydrates are preferably oxidised in order to maintain carbohydrate balance. As a consequence, fat oxidation is impaired [39]. In contrast, our study did not identify carbohydrate and fat oxidation. However, although the decrease in carbohydrate intake was not observed in our study, the previous study showing that fat oxidation was impaired after taking VCO explained the evidence of the decrease in profile lipids after taking VCO in our study.

The difference of energy intake and cholesterol intake daily among the VCO group before and after its consumption is also shown in the study. After the consumption of VCO, the amount of both energy intake and cholesterol intake considerably declines. This might be caused by the consumption of VCO has led to a decrease in the appetite of the respondents. Even though one study found that VCO which is added to breakfast among women with excess body fat suppressed less the feeling of hunger [39], another study suggests that VCO consumption may suppress the desire to eat and may affect postprandial Plasma Peptide YY among people with normal weight and especially obese male respondents [38].

This study has several strengths. First, regarding statistics, our participants were homogenous in almost all characteristics including BMI, energy intake, carbohydrate intake, cholesterol intake, ABI and lipid profiles. Second, VCO was provided as a supplement to the daily food intake of each participant. Finally, the participants were asked to maintain their usual habits during the VCO and control periods.

Some limitations should also be acknowledged. This study was only conducted with a small sample size because of the dropout rates. Generalising the findings for people outside the age range of adults, particularly those with comorbidities, should be done cautiously. In addition, this study did not control the physical activities of the participants, and this might affect the non-significant results obtained for TC, TG and ABI in the VCO group.

## 5. Conclusions

An increase in HDL level and a decrease in LDL level were found among people with DM and dyslipidaemia taking 1.2 mL/kgBW of VCO daily in three divided doses for 30 days and no harmful side effects were found. Future studies should investigate the duration and dosage of VCO in patients to decrease TG and total TC as well as the mechanism of how the VCO could decrease the lipids.

## Figures and Tables

**Figure 1 nutrients-15-00564-f001:**
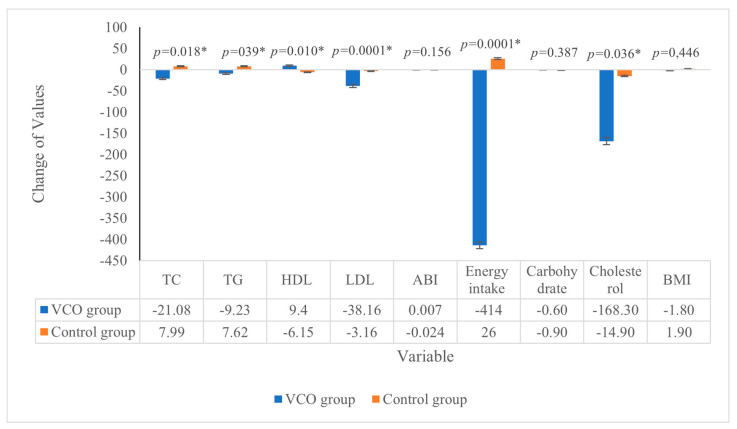
Range of different biochemistry, ABI, diets and BMI Values between VCO group and control group (Mean ± s.d.). There were significant range of differences in TC, TG, HDL, LDL, energy intake and cholesterol intake (α < 0.05; * significant different).

**Table 1 nutrients-15-00564-t001:** Baseline Characteristics of the participants who completed the study.

Baseline Characteristics	Intervention Group (*n* = 68)	Control Group (*n* = 68)	*p*-Values
	Mean (s.d.)	Mean (s.d.)	
Age (years)	63.06 (8.4)	48.00 (8.1)	<0.0001 *^,a^
Sex (% male)	53.2	52.3%	0.034 ^a^
BMI (kg/m^2^)	26.0 (3.5)	26.7 (4.1)	0.715 ^a^
Energy intakes (kilocalories/day)	1768 (346.7)	1452 (462.9)	<0.0001 *^,a^
Carbohydrates intake (% of energy)	43.3 (7.4)	45.7 (6.7)	0.525 ^b^
Cholesterol intake (mg/day)	373.6 (8.2)	298.1 (7.6)	0.528 ^b^
Total cholesterol (mg/dL)	247.69 (8.95)	231,85 (8.69)	0.167 ^a^
Triglyceride (mg/dL)	192.77 (3.27)	135.46 (3.80)	0.087 ^a^
HDL (mg/dL)	39.92 (1.44)	47.15 (2.11)	0.064 ^a^
LDL (mg/dL)	179.62 (7.75)	156.31 (6.19)	0.110 ^b^
Ankle brachial index	0.958 (0.09)	0.987 (0.08)	0.756 ^a^

^a^ Independent *t*-test. ^b^ Mann–Whitney U. * significantly different.

**Table 2 nutrients-15-00564-t002:** Outcomes of the VCO group and control group at baseline (day 1) and endpoint (day 30).

Variables	VCO Group			Control Group		
	Mean at Baseline (s.d)	Mean at End-Point (s.d)	*p*-Values	Mean at Baseline (s.d)	Mean at End-Point (s.d)	*p*-Values
Total cholesterol (mg/dL)	247.69 (8.95)	226.61 (8.05)	0.127 ^a^	231.85 (8.69)	239.84 (7.63)	0.327 ^a^
Triglyceride (mg/dL)	192.77 (3.27)	183.54 (4.96)	0.88 ^a^	135.46 (3.80)	143.08 (5.94)	0.203 ^a^
HDL (mg/dL)	39.92 (1.44)	49.32 (2.31)	0.031 ^a,^*	47.15 (2.11)	41.00 (2.67)	0.062 ^a^
LDL (mg/dL)	179.62 (7.75)	141.46 (7.15)	0.002 ^b,^*	156.31 (6.19)	153.15 (7.43)	0.797 ^b^
Ankle–brachial index	0.958 (0.09)	0.965 (0.06)	0.653 ^a^	0.987 (0.08)	0.963 (0.04)	0.627 ^a^
Energy intakes (kilocalories/day)	1768 (346.7)	1354 (257.9)	0.046 ^a^	1452 (462.9)	1478 (340.7)	0.098 ^a^
Carbohydrates intake (% of energy)	43.3 (7.4)	42.7 (7.2)	0.374 ^a^	45.7 (6.7)	44.8 (8.1)	0.326 ^b^
Cholesterol intake (mg/day)	373.6 (8.2)	205.3 (8.8)	0.023 ^b^	298.1 (7.6)	283.2 (6.9)	0.376 ^b^
BMI (kg/m^2^)	26.0 (3.5)	24.2 (4.2)	0.284 ^a^	26.7 (4.1)	28.6 (5.9)	0.198 ^a^

^a^ Dependent *t*-test. ^b^ Wilcoxon sign rank test. * significant different.

## Data Availability

The raw data and analyses conducted in this study are under restrictions due to the confidentiality requirements by ethical approval committee but can be available upon receiving a plausible request and approval from the ethical committee.

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
