# Peer review of "Virgin Coconut Oil: A Dietary Intervention for Dyslipidaemia in Patients with Diabetes Mellitus"

_nutrients, 2023, doi:10.3390/nu15030564_

Round 1

Reviewer 1 Report

Title: is it possible to speak about "therapy" concerning food supplement ? Coconut oil in not a drug that can be considered with a therapeutic effect.

L44: I do not understand what "which year" means

L49: hyperlipidaemia is not a common DM complication: as written, it can be considered that is the main characteritic. The authors should write differently.

L55: VCO is not a therapy.

L58: VCO effect is compared to simvastatin effect. Please add informations concerning bioavailability, absorption and target of polyphenols contained in VCO.

L81-82: correction: oxidase ?

L85: please describe the Friedewald's formula

L141: corection "most" and not "Most"

Figure 1: please indicate the results as mean +/- SD or SEM

L188-227: The discussion is not related with the results. Numerous informations are based on litterature and not on experimentation. There is a lack of results to assess the discussed proposition

Author Response

Dear reviewer 1,

Thank you for the correction. We revised the article one by one of your comments.

Attached is responses to your comments.

Reviewer 2 Report

Comments on “Virgin Coconut Oil: An Alternative Therapy for Dyslipidaemia 2 on Patients with Diabetes Mellitus 3”

This is very interesting that assesses the effect of virgin coconut oil (fat). After 30 days the serum lipids were measured as well as ankle–brachial 16 index (ABI). The work was done in 136 patients with diabetes (68 in each group). The work showed a decrease in LDL and an increase in HDL. But no change in total cholesterol or triglycerides, not in the ABI.

The paper is relevant and interesting, but lacks sufficient detail.

Please at a clear statistics section to the M&M section

It seems unlikely that a study of this size did not show any participant leaving the study and the number of 68 per group is not clear. Please provide clear detail on power calculation, numbers recruited and numbers finishing the study, and info on reason for leaving the study.

Table 1 lacks numbers and units. Also it would be good to add anthropometry data  and biochemistry data. Please show if any of the characteristics were different at baseline.

Please provide better information on the change in biochemical data and the exact p-value (in the abstract the change in LDL is given at p=0.002, while according to figure 1 the change is less than 0.001)? and provide test used to determine this.

According to the methods section, diets were assessed. Please show if the inclusion of VCO had an effect on the diet or not.

Figure 1 is of poor quality. I would suggest to add all individuals in the plot. And either adjust all y-axes to only cover range of the difference (like ABI) or include the range from 0 to maximum (like HLD) but do not use both approaches in one figure.

Was there a change in weight over the 30 days?

Author Response

Dear reviewer 2,

Thank you for the compliment and for the correction. We revised the article one by one of your comments.

Attached is responses to your comments.

Round 2

Reviewer 1 Report

The authors have taken into account the remarks and comments made previously